# The Influence of the Ion Implantation on the Degradation Level of the Coated Particles of Nuclear Fuel Samples

Zuzanna M. Krajewska [1,*], Tomasz Buchwald [2], Andrzej Droździel [3], Wacław Gudowski [1], Krzysztof Pyszniak [3], Tomasz Tokarski [4] and Marcin Turek [3]

[1] Laboratory for Nuclear Energy and Environmental Analyses (UZ3), National Centre for Nuclear Research, ul. Andrzeja Sołtana 7, Swierk, 05-400 Otwock, Poland
[2] Institute of Materials Research and Quantum Engineering, Poznan University of Technology, ul. Piotrowo 3, 60-965 Poznan, Poland
[3] Institute of Physics, Maria Curie-Skłodowska University, Pl. M. Curie-Skłodowskiej 5, 20-031 Lublin, Poland
[4] Academic Centre for Materials and Nanotechnology, AGH University of Science and Technology, al. A. Mickiewicza 30, 30-059 Krakow, Poland
* Correspondence: zuzanna.krajewska@ncbj.gov.pl

**Abstract:** TRIstructural ISOtropic (TRISO)-particle fuel is nuclear fuel used in high-temperature reactors. During reactor operation, partial damage may occur to the covering layers of this fuel. The authors of the publication propose an ion implantation method as a surrogate for neutron irradiation in the reactor core. This method makes it possible to reflect the damage that can be caused by irradiating samples in the reactor much faster and without having to deal with radioactive material. This paper presents an experiment on the p-TRISO samples, with a focus on the level of damage to the covering layers that could occur after 1, 3 and 5 years of neutron irradiation. The paper presents research conducted on both polished and unpolished p-TRISO fuel samples implanted with ions (Ne$^+$, He$^+$) of appropriate fluence and energy. It is necessary to determine whether the passage of time affects the occurrence of structural changes in p-TRISO fuel layers and, at the same time, whether it contributes to an increase in the probability of damage in the examined fuel material. The result of this work is confirmation that ion implantation is an efficient tool for reflecting irradiation-induced damage in the p-TRISO samples. In addition, the assumption that the sample does not need to be polished to obtain information about damage in the p-TRISO covering layers was confirmed.

**Keywords:** TRISO; irradiation; neon; helium; damage; Raman spectroscopy; passage of time

## 1. Introduction

The IV generation nuclear reactors currently are undergoing a renaissance, mainly due to the relaunched research on the TRISO (TRIstructural ISOtropic)-particle fuel. This type of nuclear fuel is a common fuel for high-temperature gas-cooled reactors (HTGRs) and can be distinguished in the form of the pebble bed and prismatic type of samples packed in a graphite matrix [1]. In both cases, the TRISO-particle fuel serves as the smallest component on which the experiments should be performed to improve the quality of the samples. During neutron irradiation in the reactor core, partial or complete damage may occur in the covering layers of the TRISO fuel. This sort of phenomenon is a mishandled effect, and ongoing work to mitigate the problem plays a key role in improving the newly produced fuel. The recirculation of the fuel in HTR reactors, such as HTR-10 and HTR-PM, takes 3 to 5 years in general [2,3]. Until now, the only way to verify the level of damage to the fuel structure was to perform examinations only after the reactor had finished operating, after extracting samples from the reactor core. The ion implantation method allows for verifying the level of damage to TRISO fuel at a significantly faster rate without the necessity to deal with activated materials. Due to that, in this experiment, authors choose such implantation fluence, which corresponds to the neutron irradiation in the reactor core of 1, 3 and 5 years.

The ion implantation method allows for quickly simulating the fuel irradiation phenomenon through the displacements per atom (dpa) parameter, which will determine the level of damage in the TRISO layer structure. The dpa is the number of point defects, meaning the average number of times an atom is displaced from the original lattice, which is widely used as an exposure parameter to evaluate the atomic-level structural damage in irradiated materials. The formula to calculate the dpa is as follows [4–7]:

$$\text{dpa} = \frac{\Phi * damage - rate * 10^8}{N} \tag{1}$$

where $\Phi$ is the fluence in ions/cm$^2$, the damage rate is in vacancies/ion/Å obtained with TRIM [8] calculations, and N is the atomic number density in atoms/cm$^3$.

A presented experiment was performed on surrogate samples of TRISO particles, the so-called p-TRISO. The p-TRISO samples are composed of the zirconium dioxide kernel, covered with Buffer porous layer, and an Inner pyrolytic carbon (IPyC) layer. These nuclear fuel samples were polished to obtain access to covering layers and implanted with specific ions and energy, which reflect the neutron irradiation fluence at the reactor core. The Raman spectroscopy method and the scanning electron microscope (SEM) were used as experimental tools to track the ongoing changes in the structure of p-TRISO layers. Since TRISO fuel layers are based on pyrolytic carbon, which is similar to graphite, it is possible to investigate the changes occurring in the spectrum after ion implantation.

In the case of Raman spectra obtained before implantation, one can notice two characteristic D-and G-bands (Figure 1—report in print) for carbon-based materials. The pristine sample exhibited a G-band, which is caused by the in-plane bond-stretching motion of sp$^2$ C atoms, and the D-band, which intensity is inversely proportional to the dimensions of crystal structures, thus the intensity is expected to increase with increasing radiation damage. The second-order Raman spectrum reveals the three main bands, such as the G*-band (2450 cm$^{-1}$), 2D-band (2680 cm$^{-1}$) and 2D'-band (3250 cm$^{-1}$), where the 2D- and 2D'-bands are the harmonics of Raman inactive fundamental modes in ordered carbons [9,10].

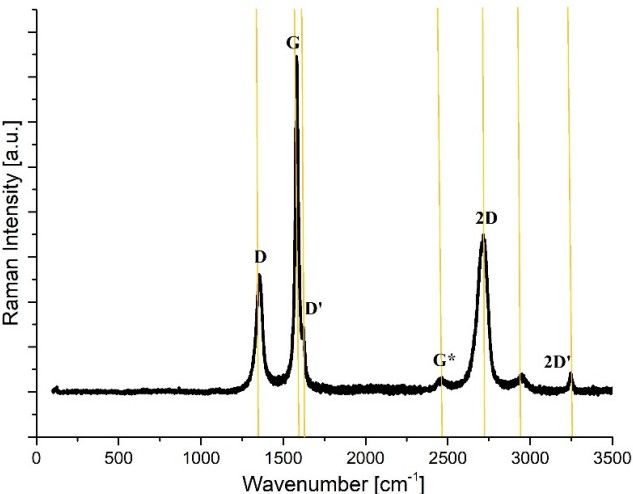

**Figure 1.** Raman spectra obtained for a pristine graphite sample.

This publication is a continuation of previous research (report in print), where neon ions were used to reflect the neutron irradiation process on analyzed p-TRISO samples. In addition to neon ions, in this work, the authors decided to use also helium ions in the experiment of ion implantation. The irradiation with lightweight He$^+$ ions creates less damage in the lattice structure of analyzed materials and, at the same time, gives good control over the implantation depth. In addition, helium is used in HTR reactors as a coolant [11–13].

In the experiment, different ion fluences were used for the ion implantation experiment. The ion fluences that reflect the 1, 3 and 5 years of neutron irradiation are presented in Table 1.

**Table 1.** Parameters used for the irradiation experiment.

| Ion Energy | Fluence [ions/cm$^2$] | DPA | Year of Neutron Irradiation |
|---|---|---|---|
| Ne$^+$ (160 keV) | $3.8 \times 10^{16}$ | 8 | 1 |
| | $1.1 \times 10^{17}$ | 23 | 3 |
| | $1.9 \times 10^{17}$ | 40 | 5 |
| He$^+$ (160 keV) | $1.05 \times 10^{18}$ | 24 | 3 |

The use of the two diagnostic methods, such as Raman spectroscopy and SEM, was driven by the fact that the results might be obtained in a fast and not expensive way, which as such, improves the quick screening methods of the TRISO samples.

## 2. Materials and Methods

### 2.1. p-TRISO

The experiment was performed on surrogate in-process coated particles. These samples were manufactured at the Centre d'Etudes Nucléaires de Grenoble (CEA, Grenoble, France) in 2001 for the TRISO fuel development, production and examination high-temperature rector project. The surrogate samples consist of zirconium dioxide (ZrO$_2$) kernel coated only with the Buffer and Inner-pyrolytic carbon (IPyC) layers. Due to the layer composition and thickness (Buffer-100 μm, IPyC-65 μm), they can not be called TRISO-particles, nor as a BISO. In this paper, as well as in the previous research [14], those samples are called predecessors to TRISO—"p-TRISO S-3" type of samples. In the experiment, two types of p-TRISO S-3 samples were used, the unpolished and polished samples (the ion polishing technique was performed to obtain the cross-sections on which further experiments may be performed; Figure 2).

**p-TRISO S-3**

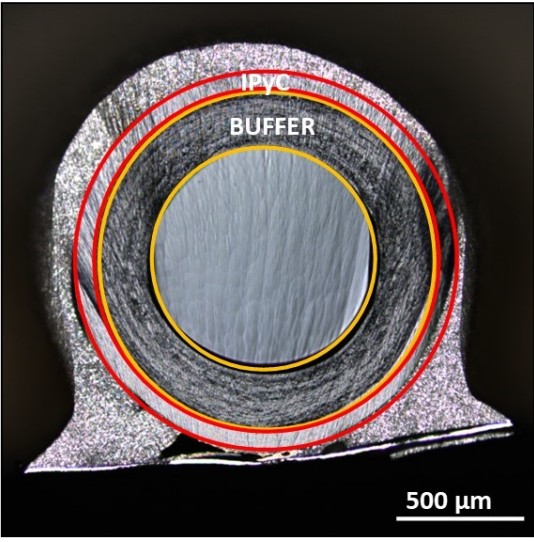

**Figure 2.** The SEM image of the cross-section of the S-3 p-TRISO sample; with the red color the IPyC layer (thickness: ~65 μm, density: 1.84 g/cm$^3$) and with yellow color of the Buffer layer (thickness: ~100 μm, density: 0.97 g/cm$^3$) were marked.

## 2.2. Methods

The ion implantation method was chosen to reflect the neutron irradiation process of the TRISO fuel in the reactor core. Ion beam irradiation on p-TRISO samples was performed using the UNIMAS ion implanter. The ion beam was produced using a universal arc discharge ion source. Several of the S-3 p-TRISO samples were attached separately to the sample holder using conducting carbon tape and irradiated at room temperature with the neon or helium ions. Because of the room temperature, the thermal annealing of defects due to the implantation process cannot be accounted for, only the primary displacement damage. Irradiation current density was in the range from 0.01 $\mu$A/cm$^2$ up to 1 $\mu$A/cm$^2$ depending on the fluence, and the pressure in the chamber was approximately $10^{-6}$ mbar. The ion implantation occurred over the entire surface of the polished p-TRISO S-3 samples. For the experiment, two types of ions were used, neon and helium. The reason for selecting these types of ions was due to the fact that these ions are light (Ne$^+$) and heavy (He$^+$) and have large penetration ranges into the analyzed structure, allowing the results to be analyzed by Raman spectroscopy.

Measurements on p-TRISO samples were carried out by an inVia Renishaw micro-Raman system, and the spectra were collected in the spectral range of 200–3200 cm$^{-1}$ by focusing on a 785 nm laser beam. The observations of the p-TRISO cross-sections were performed by FEI Versa 3D Scanning Electron Microscope.

## 3. Results

### 3.1. Verification of the Level of Damage to the p-TRISO Fuel Covering Layers Structure after 1st and 5th Year of Irradiation

The current analysis of p-TRISO particles was performed on the six samples of S-3 type p-TRISO particles to be able to perform the statistical analysis on the obtained results. Figure 3 shows the cross-section of the S-3 p-TRISO sample with average Raman spectra obtained for Buffer and IPyC layers. The layer structure of both covering p-TRISO layers changed during the implantation procedure. During the ion implantation process with neon ions (Ne$^+$), the energy of 160 keV and specified fluence, the increasing damage in Buffer and IPyC layers is observed. The ion implantation fluence applied in this experiment reflects the period of time during which samples are irradiated with neutrons in the reactor core during reactor operation. Based on performed calculations with the use of SRIM and MCB codes [8,15], it was possible to establish specific fluences of ion implantation. It was calculated that 1 year of neutron irradiation reflects the fluence of $3.8 \times 10^{16}$ ions/cm$^2$ of Ne$^+$ ions implanted and that 5 years of neutron irradiation reflects the fluence of $1.9 \times 10^{17}$ ions/cm$^2$ of Ne$^+$ ions implanted. Therefore, for S-3 samples with an averaged width of 165 $\mu$m and a layer density of 1.405 g/cm$^3$, subjected to ion implantation (Ne$^+$, 160 keV), using the detailed calculation method and taking into account ion penetration to the depth of 4257 Å in the Buffer + IPyC layers, the dpa parameter is equal 8 dpa after 1st year of irradiation ($3.8 \times 10^{16}$ ions/cm$^2$) and 40 dpa after 5th year of irradiation ($1.9 \times 10^{17}$ ions/cm$^2$).

The SEM images (Figure 3) obtained for samples type S-3 show that with increasing implantation fluence, the structure of each layer becomes more damaged. Based on previous research, one can notice that the microstructure of the low-density porous pyrocarbon (Buffer) layer becomes globular when the microstructure of high density (isotropic) Inner-pyrolytic carbon (IPyC) layer become more conical (polyhedral). This is represented by the formation of cones in the IPyC layer and circles in the Buffer layer [16]. The difference in the microstructure of both layers is due to the difference in the density of analyzed layers (Buffer—0.97 g/cm$^3$; IPyC—1.84 g/cm$^3$).

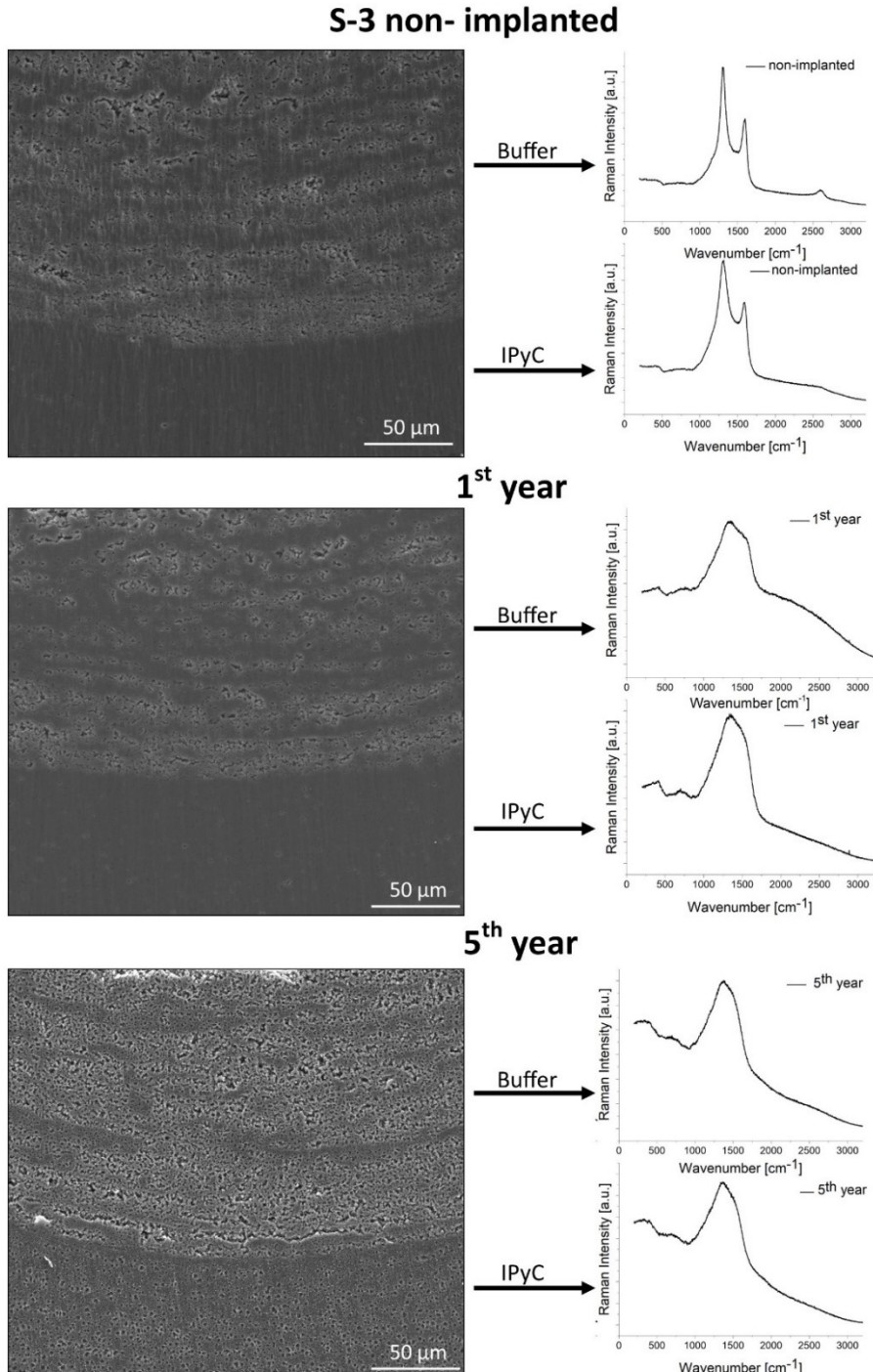

**Figure 3.** On the left-hand side—SEM image of the p-TRISO sample covering layers (Buffer, IPyC) before and after ion implantation; on the right-hand side—Raman spectra obtained for each of the analyzed layers after the ion implantation process.

Tables 2 and 3 presents the data obtained from the Raman spectra, such the position of D- and G-bands, the width of the D- and G-bands and the intensity ratio of the D-Raman band and G-Raman band, as well as the area of ID/IG.

**Table 2.** The average values obtained from six samples of S-3 p-TRISO buffer layer.

| Fluence | Position | | Width | | ID/IG | ID/IG Area |
|---|---|---|---|---|---|---|
| | D-Band | G-Band | D-Band | G-Band | | |
| before implantation | 1309 ± 0.4 | 1583.8 ± 1.1 | 140.5 ± 3.1 | 106 ± 2.3 | 1.69 ± 0.03 | 2.05 ± 0.03 |
| 1st year of irradiation | 1348.7 ± 7.1 | 1551.7 ± 12.6 | 461.8 ± 26.9 | 133.8 ± 19.6 | 3.85 ± 0.58 | 18.38 ± 5.0 |
| 5th year of irradiation | 1373.3 ± 5.4 | 1537.7 ± 3.6 | 432.4 ± 16.1 | 153.9 ± 10.5 | 4.85 ± 0.77 | 18.74 ± 5.44 |

**Table 3.** The average values obtained from six samples of S-3 p-TRISO IPyC layer.

| Fluence | Position | | Width | | ID/IG | ID/IG Area |
|---|---|---|---|---|---|---|
| | D-Band | G-Band | D-Band | G-Band | | |
| before implantation | 1315.6 ± 1.5 | 1578.2 ± 0.6 | 243.3 ± 3.2 | 129 ± 2.5 | 1.56 ± 0.06 | 3.34 ± 0.25 |
| 1st year of irradiation | 1332.2 ± 8.6 | 1529.4 ± 8.2 | 426.5 ± 24.8 | 177.6 ± 16.2 | 2.85 ± 0.56 | 9.05 ± 3.9 |
| 5th year of irradiation | 1371.7 ± 5.5 | 1544.9 ± 4.5 | 441.0 ± 18.6 | 147.1 ± 14.1 | 5.29 ± 0.98 | 21.91 ± 8.0 |

The typical Raman spectra of carbon-based materials contain the D-band at the position around 1310 $cm^{-1}$ and the G-band around 1580 $cm^{-1}$. Based on the literature, "G" comes from graphite and "D" comes from disorder. Each noticeable change from these values means creating defects in the analyzed structure. In the beginning, when the layers of the p-TRISO sample were not implanted, two separate bands were noticed. With the increasing fluence, these two D- and G-bands start to create one single band, which is due to the fact that the dislocation process starts to occur at investigated layer structures. The reason may be apparent in the increasing values of the width parameter. The ratio of intensities of the bands of the disorder-induced D band and the first-order graphite G band (ID/IG) is crucial to estimating the structural disorder. In both cases, on examined layers, one can observe that the intensity ratio ID/IG increase with the time of sample implantation.

Raman spectroscopy measurement can also be used to prepare the surface maps of the analyzed samples. During the measurement, Raman spectra were collected every 1 μm step. The goal of the measurement was to obtain a large amount of data from the selected field. The surface maps are shown in Figure 4. This figure presents the average six maps obtained for S-3 p-TRISO particles. Maps are divided according to the time of implantation, such as before implantation, 1st year of implantation and 5th year of implantation. Moreover, one can notice two columns, maps for the Buffer layer and maps obtained for the IPyC layer. In each case, the map shows the concentration of the specified layer. The highest concentration is marked with red color, which means 100% of a specified layer in the region. Based on that, one can notice that in each case, the map shows the Buffer layer at the bottom and the IPyC layer at the top of the selected region (30 × 50 μm). In the case of the maps obtained before S-3 samples implantation, the junction of Buffer and IPyC layers was at the level of 15 μm. With the increasing time of implantation, the Buffer–IPyC junction shift to the level of 8 μm. Such behavior indicates significant damage at the interface of the Buffer–IPyC layers, which may provide the creation of a gap between those layers.

*3.2. Verification of the Level of Damage to the p-TRISO Fuel Covering Layers Structure after 3rd Year of Irradiation*

To be able to consider the ion implantation method as a fast screening tool for newly produced TRISO particles before inserting them into the reactor core, S-3 samples that had not been polished were subjected to ion implantation. Accordingly, this part of the experiment was carried out on p-TRISO S-3 spheres in such a way that the ion beam fell on the outer surface of the sample, in this case, on the IPyC layer (Figure 5). The p-TRISO samples were divided into two groups, where one group was implanted with $Ne^+$ ions to continue performing research and the second group of samples was implanted with $He^+$ ions [17,18]. The reason for choosing helium ions comes from the fact that the ion

penetration of these ions is deeper (8361 Å) compared to the neon ions (4257 Å). In addition, helium is a chemical component that serves as a coolant in HTR reactors.

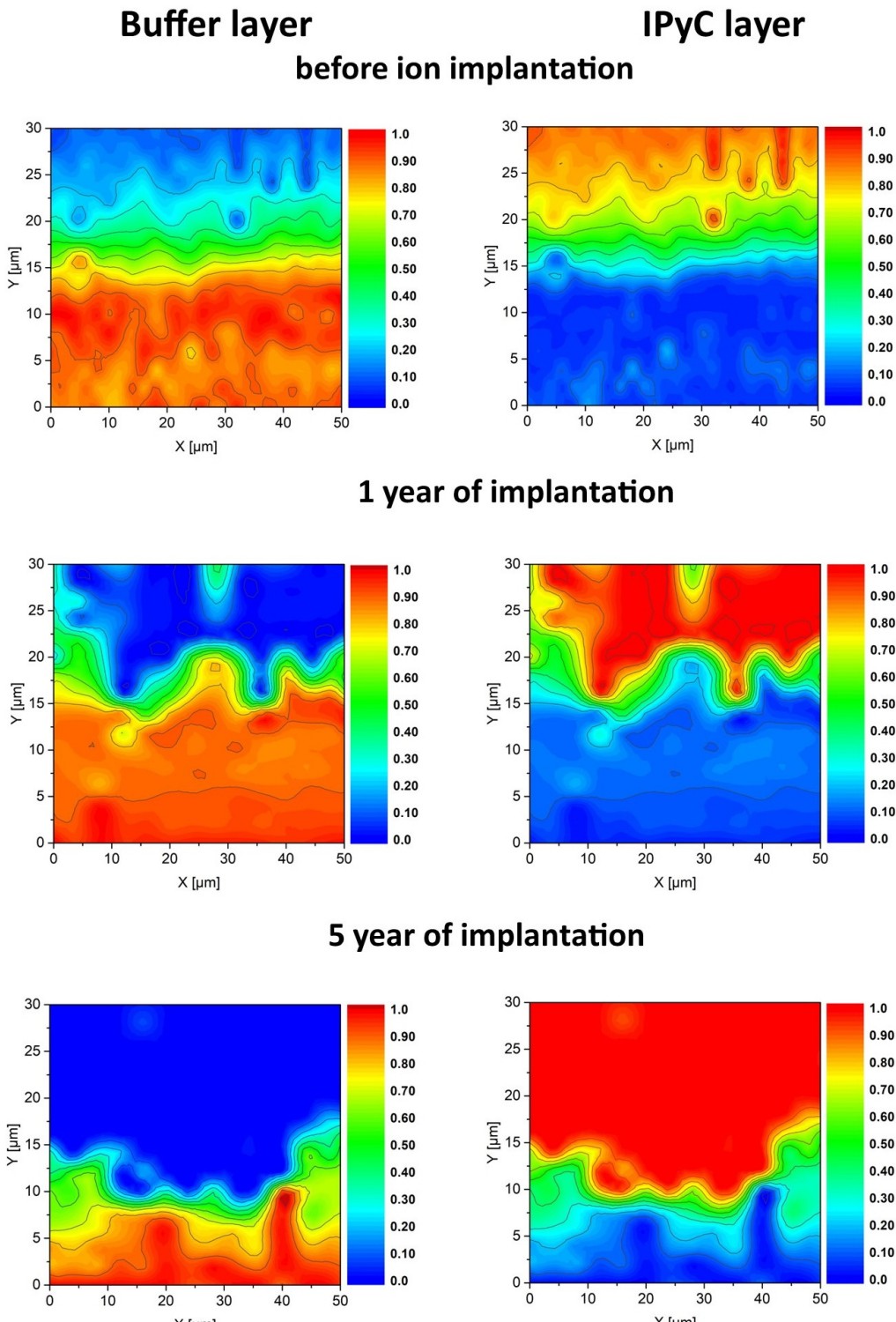

**Figure 4.** Raman maps of the averaged S-3 p-TRISO particles.

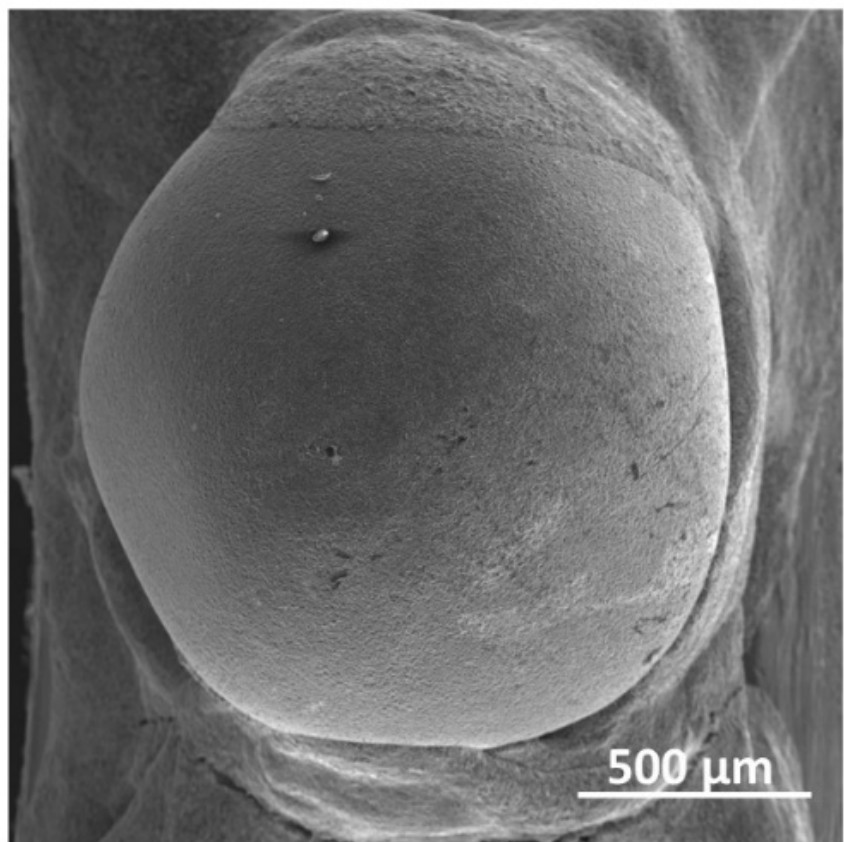

**Figure 5.** SEM image of the non-polished p-TRISO particle sphere.

The experiment on non-polished samples assumed that the energy of implantation would remain the same as in the previous experiment (i.e., 160 keV), but the fluence of ion implantation will change for one, which reflects 3 years of neutron irradiation process in the reactor core. Based on the SRIM and MCB calculations, it was concluded that 3 years of neutron irradiation reflects the fluence of $1.1 \times 10^{17}$ ions/cm$^2$ of Ne$^+$ ions implanted and that 3 years of neutron irradiation reflect the fluence of $1.05 \times 10^{18}$ ions/cm$^2$ of He$^+$ ions implanted. The dpa after 3rd year of irradiation is equal 23 dpa ($1.1 \times 10^{17}$ Ne-ions/cm$^2$) and 24 dpa ($1.05 \times 10^{18}$ He-ions/cm$^2$). Therefore, for p-TRISO samples with a layer density of 1.405 g/cm$^3$, subjected to ion implantation (He$^+$, 160 keV), using the detailed calculation method, the ion penetration depth is equal to 1.06 μm.

Based on Figure 6, one can observe the structure deformation due to the ion implantation. The magnification of selected regions obtained for both cases shows globular formation. The PyC layer is composed of microcrystalline structures about 500 nm in diameter, which contains domains or crystallites in the size of about 2–3 nm. These domains may increase in size from about 2 to 5 nm under irradiation and heat treatments at a temperature of 1800 °C [19–21]. The results obtained from Raman spectra are shown in Table 3. In both cases of analyzed ions, one can observe that the D-band position increase when the G-band position decrease, which is related to the formation of a disordered structure. For the rest of the selected parameters, it is observed that due to the 3 years of implantation, the values of width for D-and G-band, as well as the intensity ratio ID/IG and are increased over time. What should be noted is that the increment is bigger for He$^+$ irradiated ions compared to that irradiated with Ne$^+$ ions.

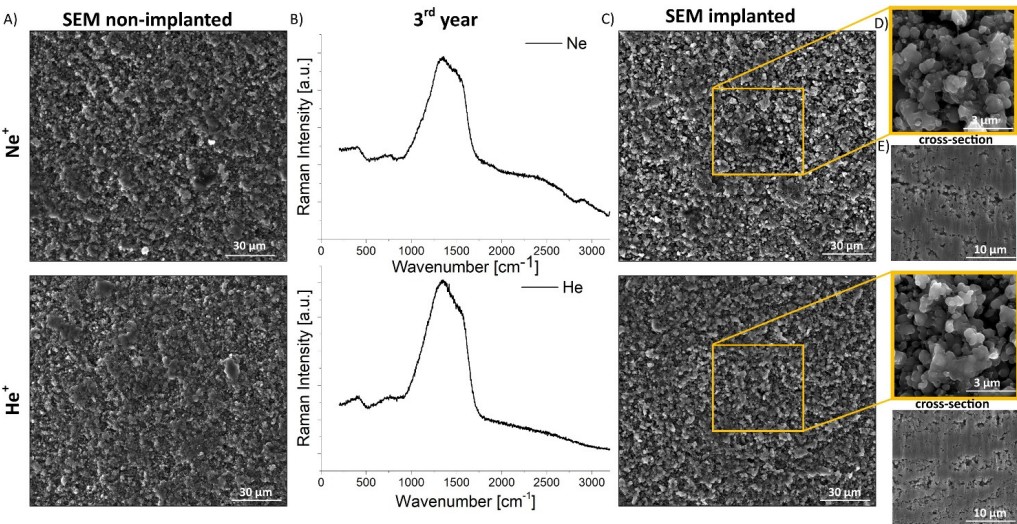

**Figure 6.** (**A**) SEM images of non-implanted surface of S-3 particle, (**B**) Raman spectra obtained for the Ne$^+$ and He$^+$ implanted samples with the fluence equivalent to 3 years of irradiation, (**C**) SEM images of implanted S-3 samples—3rd year of implantation with Ne$^+$ and He$^+$ ions; (**D**) SEM magnification of selected region; (**E**) the cross-section of S-3 implanted sample.

This analysis (Table 4) was performed on non-polished samples and shows that it is possible to analyze the surface degradation on p-TRISO particles. To confirm this assumption, after Raman spectroscopy measurements, samples that were implanted with Ne$^+$ and He$^+$ ions were subjected to the ion polishing process. After polishing, samples were again verified by Raman spectroscope; results are presented in Table 5.

**Table 4.** S-3 outer surface implantation.

| Fluence | Position | | Width | | ID/IG | ID/IG Area |
|---|---|---|---|---|---|---|
| | D-Band | G-Band | D-Band | G-Band | | |
| before implantation | 1315.6 ± 0.7 | 1587.9 ± 0.9 | 228.3 ± 14.2 | 115.1 ± 7.4 | 1.68 ± 0.06 | 2.68 ± 0.66 |
| 3rd year of Ne$^+$ irradiation | 1347.2 ± 10.4 | 1556.8 ± 14.1 | 445.1 ± 52.6 | 130.9 ± 22.6 | 3.43 ± 0.95 | 17.0 ± 8.79 |
| 3rd year of He$^+$ irradiation | 1340.4 ± 6.5 | 1567.6 ± 4.5 | 483.6 ± 45.7 | 137.2 ± 12.8 | 3.7 ± 0.82 | 17.39 ± 6.95 |

**Table 5.** S-3 cross-section after irradiation.

| Fluence | Position | | Width | | ID/IG | ID/IG Area |
|---|---|---|---|---|---|---|
| | D-Band | G-Band | D-Band | G-Band | | |
| before implantation | 1309.9 ± 1.7 | 1585.3 ± 2.7 | 188.6 ± 18.4 | 105.5 ± 7.8 | 1.69 ± 0.08 | 3.16 ± 0.63 |
| 3rd year of Ne$^+$ irradiation | 1332.5 ± 3.7 | 1573 ± 3.2 | 334.9 ± 15.1 | 145.5 ± 5.5 | 2.69 ± 0.17 | 8.97 ± 0.89 |
| 3rd year of He$^+$ irradiation | 1323.8 ± 2.5 | 1578.3 ± 2.2 | 332.8 ± 22.6 | 129.7 ± 11.1 | 2.64 ± 0.19 | 9.59 ± 1.73 |

Based on obtained results (Table 4), one can notice significant changes in the layer structure before and after sample implantation, which occur at the cross-section of implanted samples. Regardless of whether the sample was first ion-implanted and then polished or the other way, first polished and then ion-implanted, in both situations, we hold the same effect of detecting implantation-induced damage in the layers covering p-TRISO samples. It should be kept in mind that, in this case, only the IPyC layer is analyzed, as this is the outer layer of the TRISO sample under investigation. Observation of visible changes in the structure of this layer confirms the depth of penetration of ions into its structure. For these reasons, the values obtained for the buffer layer will be unchanged since the depth of ion penetration was too small to reach this particular layer. In order to be able to determine the

damage created in the Buffer layer on unpolished p-TRISO samples, it would be necessary to increase the implantation energy and thus extend the depth of penetration of ions into the layer structures in order to find them in both the IPyC layer and the Buffer layer.

## 4. Discussion

The first purpose of this research was to examine the p-TRISO particles in terms of the damage that occurs in the covering layers. Research techniques such as Raman spectroscopy and SEM were used for the investigation. The second purpose was to present fast and not expensive measurement devices for quick verification of the pyrocarbon layers; due to that, the research was limited only to these two diagnostic methods. The analysis was done on the S-3 type p-TRISO samples. Based on performed experiments, it can be concluded that the ion implantation method is a significant method to reflect the neutron irradiation process in the reactor core. Due to the ion implantation with fluence that reflects 1, 3 and 5 years of TRISO- particle irradiation, one can observe the partial damage that may occur during HTR reactor operation.

This experiment was devoted to measuring how the passage of time influences the degradation of the layers in the tested TRISO particles. As a result, the level of damage was presented due to the dpa parameter, which is equal to 8 dpa—after 1st year of irradiation, 23 dpa—after 3rd year of irradiation and 40 dpa—after 5th year of irradiation. Based on the obtained results, one can notice that the passage of time affects the occurrence of structural changes in the TRISO fuel layers and, at the same time, it contributes to an increase in the probability of damage in the examined fuel material.

One has to bear in mind that this research did not take into account the graphite matrix in which the TRISO particles are immersed, creating 6-cm pebbles or 4-cm pellets. Due to that, these assumptions may happen only on the TRISO particles without any additional cover, but still, the obtained results on the degradation level should be taken into account for the new production of fresh TRISO particles. One has to bear in mind also the fact that analyzed p-TRISO particles consist only of two covering layers (they do not have the most protective silicon carbide (SiC) layer) and are 22 years old, which as such, may induce faster degradation.

Nevertheless, this publication proves that for a fast screening method of a stored or a newly manufactured fuel, the ion implantation technique may be used to verify the quality of TRISO particles before entering the reactor core. In addition, the experiment conducted on unpolished samples confirms the assumption that implantation on the spheres is sufficient for the fuel screening method without the necessity of polishing the samples to access the inner layers. To obtain data on the level of damage to the inner layers, the energy with which the implantation process would be performed should be increased. This research step will be confirmed in subsequent studies.

**Author Contributions:** Conceptualization, Z.M.K.; methodology, Z.M.K.; validation, Z.M.K., T.B., A.D., W.G., K.P., T.T. and M.T.; investigation, T.B., A.D., K.P., T.T. and M.T.; resources, Z.M.K.; writing—original draft preparation, Z.M.K.; writing—review and editing, T.B., A.D., W.G., K.P., T.T. and M.T.; visualization, Z.M.K.; supervision, W.G.; project administration, Z.M.K. All authors have read and agreed to the published version of the manuscript.

**Funding:** The work was supported by the Polish National Research and Development Center (NCBR) project "New Reactor Concepts and Safety Analyses for the Polish Nuclear Energy Program", POWR.03.02.00-00-I005/17 (years 2018–2023). Raman spectroscopy analysis was carried out with financial support from the Ministry of Education and Science in Poland, 0511/SBAD/2351.

**Institutional Review Board Statement:** Not applicable.

**Informed Consent Statement:** Not applicable.

**Data Availability Statement:** The data of this study are available from the corresponding author upon reasonable request.

**Acknowledgments:** The experiments were conducted thanks to the collaboration with Michael A. Fütterer (European Commission—Joint Research Center). We gladly acknowledge his invaluable support.

**Conflicts of Interest:** The authors declare no conflict of interest. The funders had no role in the design of the study; in the collection, analyses, or interpretation of data; in the writing of the manuscript; or in the decision to publish the results.

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
