# Peer review of "The Influence of the Ion Implantation on the Degradation Level of the Coated Particles of Nuclear Fuel Samples"

_coatings, doi:10.3390/coatings13030556_

Round 1
Reviewer 1 Report
The manuscript entitled “The Influence of the Ion Implantation on the Degradation level of the Coated Particles of Nuclear Fuel Samples” by Zuzanna M. Krajewska etc., explored using ion implantation as a effective tool to reflect neutron-irradiation-induced damage in the p-TRISO samples. This would be a shortcut way to access the actual degradation of covering layer of nuclear fuel particles in nuclear reactor.
The manuscript is well written, the experimental are well carried and the data are presented clearly. I think the manuscript would be in favour of the readers of Coatings. However, I highly suggest, before accept for publishing in Coaings, there are some experiment data are highly suggested to be further provided in the manuscript .
1, The manuscript only presented the influence of ion implantation time at constant ion energy and flux density. It’s recommended the influence of different ion energy and flux density should be investigated, this would be beneficial to comprehensive understanding the actual irradiation-induced damage in the p-TRISO samples.
2, Only Ramman spectra and SEM images are not enough to represent the microstructure changes after irradiation. It’s highly recommended more detailed structure data be presented, such as pore size, distribution of pore size, and the density.
3, It’s strongly recommended that some mechanical properties should be presented, such as micro-hardness, and so on. This would be beneficial to learn about the influence of irradiation on the properties of coating layers.
Reviewer 2 Report
This paper aims to examine the p-TRISO particles in terms of the damage
that occurs in the covering layers, and the analysis is useful. A few
suggestions are as follows?
1. Are the data in Fig. 1 derived by the authors or from the previous literatures? If the data are from previous research, please supplement the reference.
2. Present study just adopted SEM and Raman to conduct the research. Are there other good methods to develop such research? Please introduce the advantages of using SEM and Raman to do present research.
Reviewer 3 Report
The authors claim they "propose the use of an ion implantation method that can rapidly assess the quality of the fuel before placing it in the reactor core." The result they present only show that they can characterize changes in Raman spectrum of p-Triso ion implanted samples. To achieve the goal they claim for, they should:
- review from literature the behavior of TRiSO coating in reactor condition. The temperature of irradiation, and Wigner effect should be commented.
- review from literature the behavior of graphite under irradiation (neutron and ion) and especially how its Raman spectrum is modified ( see for example Carbon Volume 95, December 2015, Pages 364-373)
From these reviews, they could then explain how the protocol they propose is appropriate for TRISO coating characterization. The argument they gave in the second sentence of discussion (line 256): fast and not expensive method can be justified but it is not enough for a scientific sounded proof.
Miscellaneous:
The description of dpa calculation could be presented in tables. The manner by which the dpa are calculated for TRISO case should be detailed.
The manner, by which the Raman mapping of figure 4 was obtained, should be detailed.
Sentence at end of page 3 and beginning of page 4 (line 109-119) is incorrect
Author Response
Dear Reviewer,
the answers are submitted in the attached file.
Kind regards,
Zuzanna Krajewska

Round 2
Reviewer 1 Report
The revised manuscript is well written, reasonable analysis and discussion are presented. These would be interesting to the readers of Coatings, I recommend the revised manuscript to be published.
Author Response
Thank you very much for your opinion.
Kind regards,
Zuzanna Krajewska
Reviewer 3 Report
The authors answered my comments.